# Change-Point Detection for Multi-Way Tensor-Based Frameworks

**DOI:** 10.3390/e25040552

**Published:** 2023-03-23

**Authors:** Shanshan Qin, Ge Zhou, Yuehua Wu

**Affiliations:** 1School of Statistics, Tianjin University of Finance and Economics, Tianjin 300222, China; 2Department of Mathematics and Statistics, York University, Toronto, ON M3J 1P3, Canada

**Keywords:** tensor, image, change-point, maximum edge weight, histogram-based edge weight

## Abstract

Graph-based change-point detection methods are often applied due to their advantages for using high-dimensional data. Most applications focus on extracting effective information of objects while ignoring their main features. However, in some applications, one may be interested in detecting objects with different features, such as color. Therefore, we propose a general graph-based change-point detection method under the multi-way tensor framework, aimed at detecting objects with different features that change in the distribution of one or more slices. Furthermore, considering that recorded tensor sequences may be vulnerable to natural disturbances, such as lighting in images or videos, we propose an improved method incorporating histogram equalization techniques to improve detection efficiency. Finally, through simulations and real data analysis, we show that the proposed methods achieve higher efficiency in detecting change-points.

## 1. Introduction

Videos or images are common in real life. Examples include images of traffic lights to regulate traffic, videos recorded by electronic traffic cameras to track vehicles that violate regulations, and webcams in smart agriculture to monitor the ripening stage of vegetables or fruits and automatically estimate the harvest time. Videos or images include huge amounts of information, but not all are useful. Therefore, extracting useful information from videos or a sequence of images, which can be regarded as a change-point detection problem, is important. The authors of [1] proposed a distribution-free, consistent graph-based change-point detection method for recorded videos tracking cell division. After that, a non-Euclidean graph-based change-point test was proposed by [2] as an effective way to reduce random interference, and applied to detecting both landing and departure times in a sequence of bees’ flower visitation data.

Digital videos or images are special cases of multi-way tensors. A *r*-way tensor can be expressed as $X\in R^{p_{1}\times \cdots \times p_{r}}$, where R denotes the real number line and $p_{\ell }$ represents the dimension of mode-*ℓ* of X, ℓ=1,…,r. The elements of the tensor X can be accessed using *r* indices as in $x_{i_{1},\ldots ,i_{r}}$, varying $i_{\ell }$ from 1 to $p_{\ell }$, ℓ=1,…,r. A slice refers to a tensor subarray with one fixed index. For example, $X_{i_{r}}=\left\{x_{\cdot ,\ldots ,\cdot ,r}\right\}$ is defined to be the $i_{r}$-th slice of X along mode *r* (abbreviated as the $i_{r}$-th mode-*r* slice), $i_{r}=1,\ldots ,p_{r}$. Thus, X contains *r* different sets of slices. In modern statistical applications, tensors are commonly encountered in many fields. A color image can be represented as a three-way tensor, i.e., $R^{p_{1}\times p_{2}\times 3}$, where $\left\{x_{\cdot ,\cdot ,i_{3}}\right\}$, $i_{3}=1,2,3$, are mode-3 slices, the combination of which determines the color of an image. Nevertheless, both [1,2] converted three-way tensors to one-way tensors by taking the average of R,G,B mode-3 slices, which neglects the object’s color information.

In real life, one may also be interested in detecting targets with different colors. For example, detecting changes in the color of traffic lights may help daltonians reduce the limitations of visual impairment and be able to pass through traffic lights normally. Figure 1 shows a sequence of traffic light images containing different colors, such as green (t=1,…,10), yellow (t=11,…,14), and red (t=15,…,27). Traffic light colors are dominated by R,G,B mode-3 slices, not their average. In addition, detecting a change in the color of fruits or vegetables may help monitor their maturity stages and estimate their harvest time, which is important for building automated harvesting systems. For example, the ripening process of tomato fruits can be categorized into three stages: mature green (MG), breaker (BR), and light red (LR) [3,4]. Figure 2 displays 26 frames of these three stages, i.e., MG (t=1,…,8), BR (t=9,…,19), and LR (t=20,…,26), which are extracted from a video that recorded the tomato fruits’ maturation process from MG to LR. However, the pixel values of images or video frames are sensitive to light intensity. Figure 3 presents the same sequence of 26 frames, but the light intensity changes at t=5,10,17,21. All these disturbances make change-point detection more challenging.

Inspired by the above examples, we were interested in developing a change-point detection method for the multi-way tensor-based framework, which we present in this paper. Suppose an observation of an object can be represented as a sequence of *r*-way tensors. The features of an object are dominated by one or a few slices. One may encounter a situation where the distribution of one or some mode-*ℓ* slices suddenly changes, ℓ=1,…,r, say, mode-*r* slices. If so, taking the average over all mode-*r* slices may mask changes, such that some features of the object are no longer clearly visible. Following the graph-based change-point detection method in [2], we propose two detection methods based on the multi-way tensor framework. One is directly based on the elements of the slices, and the other is based on the histograms of the slices to reduce the influence of natural disturbances, such as lighting, in the image. Change-points occur when the value of some important dominant slices change suddenly. Without loss of generality, we take a three-way tensor as an example to intuitively illustrate the performance of the proposed method. We show through simulations and real-data analysis that the proposed methods are more capable than the method in [2] regardless of whether the sequence of tensors is affected by natural disturbances.

The main contributions of this study are summarized. First, to the best of our knowledge, few studies have addressed the problem of identifying temporal change-points in sequences of tensors. Therefore, we propose a graph-based change-point detection method to identify temporal changes under the framework of multi-way tensors. Second, we propose maximal and histogram-based edge weights to build graphs. The latter aims to downplay or remove natural distractions, such as lighting, in an image. Third, we conduct simulation studies in different scenarios with equal or unequal variances of the slices, with or without natural disturbances. Our simulation results demonstrate the robustness and efficiency of the proposed methods in change-point detection. Finally, our proposed methods are suitable for, but not limited to, detecting color changes of traffic lights and monitoring maturity stage changes in smart agriculture.

## 2. Methodology

### 2.1. Model Setup

Let $X_{t}$ be a sequence of *r*-way tensors, i.e., $X_{t}\in R^{p_{1}\times \cdots \times p_{r}}$, where $p_{\ell }$, ℓ=1,…,r, are positive integers, and t=1,…,T. $X_{t}$ contains $p_{\ell }$ mode-*ℓ* slices, denoted as $X_{i_{\ell },t}^{\left(\ell \right)}\in R^{p_{1}\times \cdots \times p_{\ell -1}\times p_{\ell +1}\times \cdots \times p_{r}}$, where the superscript *ℓ* denotes different modes, ℓ=1,…,r, and the subscript $i_{\ell }$ denotes the $i_{\ell }$-th slice along mode *ℓ*, $i_{\ell }=1,\ldots ,p_{\ell }$. For the collection of tensors, assume that a time point $t^{*}$ divides the observations into two segments such that $X_{t}$ in two segments are significantly different, whereby the mode-*ℓ* slices characterize the difference. Then $t^{*}$ is a change-point. For example, the color of a colorful image is dominated by mode-3 slices. A change of any mode-3 slice will result in a change in the color of this image. Generally, for the mode-*ℓ* slices, denote $S=\{1,\ldots ,p_{\ell }\}$. Let $T_{t}^{\left(\ell \right)}$ and $K_{t}^{\left(\ell \right)}$ be subsets of S for $t=1,\ldots ,t^{*}$ and $t=t^{*}+1,\ldots ,T$, respectively. Note that $K_{t}^{\left(\ell \right)}=\emptyset$ for $t=1,\ldots ,t^{*}$. Without loss of generality, make ℓ=r. Define $j=\left\{j_{1},\ldots ,j_{r-1}\right\}\in R^{r-1}$, an index label vector, where $j_{\ell }=1,\ldots ,p_{\ell }$ with ℓ=1,…,r−1. Denote $J=\left\{1,\ldots ,p_{1}\right\}\times \cdots \times \left\{1,\ldots ,p_{r-1}\right\}$. Let $x_{j,m,t}^{\left(r\right)}$ be the elements of $X_{m,t}^{\left(r\right)}$ at time *t*, j∈J and $m\in \{1,\ldots ,p_{r}\}$. Suppose that the object area of these slices is represented as $J_{1,t}^{\left(r\right)}\subset J$ at time *t*. We use the following model to illustrate the time series with a change-point at time $t^{*}$:(1)$$x_{j,m,t}^{\left(r\right)}=\left\{\begin{array}{cc} \mu_{m,1}^{\left(r\right)}+\sigma_{m,1}^{\left(r\right)}\epsilon , & j\in J_{1,t}^{\left(r\right)},\;m\in T_{t}^{\left(r\right)},\;m\notin K_{t}^{\left(r\right)},\\ \mu_{m,2}^{\left(r\right)}+\sigma_{m,2}^{\left(r\right)}\epsilon , & j\in J_{1,t}^{\left(r\right)},\;m\in K_{t}^{\left(r\right)}, \end{array}\right.$$
where $\mu_{m,i}^{\left(r\right)},\;\sigma_{m,i}^{\left(r\right)},\;i=1,2$, are unknown parameters, ϵ is an independently identically distributed (iid) random error, with a mean of zero and a variance of one. The first part in model (Equation 1) corresponds to the mode-*r* slices with no change (initial state). The second part shows significant changes among these mode-*r* slices for $t>t^{*}$. For simplicity, we drop the suffix *r* in (Equation 1). Then, model (Equation 1) becomes
(2)$$x_{j,m,t}=\left\{\begin{array}{cc} \mu_{m,1}+\sigma_{m,1}\epsilon , & j\in J_{1,t},\;m\in T_{t},\;m\notin K_{t},\\ \mu_{m,2}+\sigma_{m,2}\epsilon , & j\in J_{1,t},\;m\in K_{t}. \end{array}\right.$$

If there exists a change-point $t^{*}$, similar to [2], detecting the change-point of (Equation 1) is equivalent to testing the following hypothesis:(3)$$H_{0}:K_{t}=\emptyset ,t=1,\ldots ,T\;\;\mathrm{vs}\;\;\;H_{a}:\left\{\begin{array}{cc} K_{t}=\emptyset , & t=1,\ldots ,t^{*},\\ K_{t}\neq \emptyset , & t=t^{*}+1,\ldots ,T. \end{array}\right.$$
Suppose that a time point *t* divides the observations into two segments, ‘past’ (≤t) and ‘future’ (>t), t=1,…,T. In light of [2,5], we define the number of edges that connect points from the ‘past’ and ‘future’ segments over a graph G, for any *t*, and $\left(t_{1},t_{2}\right)\in E\left(G\left(\omega \right)\right)$,
(4)$$C_{t}^{G(\omega )}=\sum_{\left(t_{1},t_{2}\right)\in E\left(G\left(\omega \right)\right)}I\left\{I\left\{t_{1}>t\right\}\neq I\left\{t_{2}>t\right\}\right\},$$
where I{A} is an indicator function that is equal to 1 if *A* is true, and 0 otherwise; and G is a connected, edge-weighted undirected graph with vertex set {1,…,T} and edge set E(G). Many ways can be used to construct such a graph, such as minimum spanning tree (MST), minimum distance pairing (MDP), and shortest Hamiltonian path (SHP) [1,2,5]. Nevertheless, not all of them perform well in a graph-based hypothesis test. As is shown in [1], a test based on MST performed poorly in terms of power for a large dimension *d*. Moreover, Chen and Zhang [5] argue that MDP achieves the desirable property of being truly distribution-free. However, Biswas et al. [6] show that the generalized run test using the SHP is distribution-free and consistent as a dimension *d* goes to infinity. In our study, we thus use the graph G that is obtained using SHP, denoted as SHP(ω). Small values of $C_{t}^{G(\omega )}$ are against the null hypothesis $H_{0}$ in (Equation 3), which is equivalent to constructing the graph that attains the minimum of the sum of edge weights among all of the spanning paths. Therefore, it is key to establish suitable edge weights $\omega_{t_{1},t_{2}}$ between any two nodes $t_{1},t_{2}$ in the construction of graph G. We propose two edge weights, given in (Equation 7) or (Equation 8) of the following two subsections.

To test the null hypothesis, an SHP-based test was proposed in [1], with the test statistic
(5)$$S_{T}^{\mathrm{SHP}(\omega )}=\frac{1}{T-1}\sum_{t=1}^{T-1}{\left(Z_{t}^{\mathrm{SHP}(\omega )}\right)}^{2},$$
where $Z_{t}^{\mathrm{SHP}(\omega )}$ is the standardization of $C_{t}^{\mathrm{SHP}(\omega )}$, $Z_{t}^{\mathrm{SHP}(\omega )}=\left(C_{t}^{\mathrm{SHP}(\omega )}-E_{0}\right)/\sqrt{{\mathrm{var}}_{0}}$, $E_{0}=2t\left(T-t\right)/T$; and ${\mathrm{var}}_{0}=2t\left(T-t\right)\left\{2t\left(T-t\right)-T\right\}/\left(T^{3}-T^{2}\right)$. We reject the null hypothesis at the significant level α if the statistic $S_{T}^{\mathrm{SHP}(\omega )}$ is larger than the simulated critical value. When rejecting $H_{0}$, the change-point estimate is given by
(6)$$\arg\min_{1\leq t\leq T}C_{t}^{\mathrm{SHP}(\omega )}/\left\{t\left(T-t\right)\right\}.$$
Under the null hypothesis, the distribution of $S_{T}^{\mathrm{SHP}(\omega )}$ is distribution-free. The critical values are calculated via the permutation method with 100,000 replications based on various *d* and significance levels α. The details are given in the Appendix A.

### 2.2. Maximum Edge Weight

As mentioned earlier, it is possible to encounter a mode-*r* slice change, such as the first mode-*r* slice. Taking the average of all mode-*r* slices may mask the importance of the first slice and thus the variation in important features of the object. For example, when a study focuses on detecting time points of color change in a sequence of images, averaging three mode-3 slices removes the color features of these images. Undoubtedly, the method of [2] will fail to detect it. Inspired by the problem of detecting color changes, we thus propose maximum edge weights to enforce the importance of considering the mode-*r* slices.

Let E(G) be a set of edges of the undirected edge-weighted graph, the nodes of which are {1,⋯,T}. The maximum edge weights of nodes $t_{1},t_{2}$ are defined as
(7)$$\omega_{t_{1},t_{2}}=\max_{m\in \{1,\ldots ,p_{r}\}}\left|\sum_{j\in J}\left(x_{j,m,t_{1}}-x_{j,m,t_{2}}\right)\right|,$$
where j∈J means that $j_{\ell }\in \left\{1,\ldots ,p_{\ell }\right\}$ for ℓ=1,…,r−1. The methods based on the maximum edge weights and the non-Euclidean edge weights of [2] are named Maxima-SHP and nonE-SHP, respectively.

### 2.3. Histogram-Based Edge Weight

Both the nonE-SHP and Maxima-SHP are directly based on the differences between tensors’ elements, which can easily be affected by some natural factors. For example, the pixel values of the video’s frames are sensitive to light intensity, which makes the video’s frames or images either over- or under-exposed, resulting in a narrow range of intensity values of pixels. In such a case, using the method of nonE-SHP or Maxima-SHP directly may fail to detect the underlying true change-points. Histogram equalization is useful in images where the background and foreground are both light and dark, by effectively spreading highly populated intensity values. Motivated by the histogram equalization, we propose a new weight based on the histograms of all elements of the tensors rather than all elements directly. For the tensor $X_{m,t}$, let $f_{m,h_{i},t}$ be the empirical probability of occurrence of intensity level $h_{i}$ in the *m*-th mode-*r* slice ($m=1,\ldots ,p_{r}$), where $h_{i}$, i=1,…,L, denote the intensity levels, and *L* is the number of possible intensity levels (e.g., *L* = 256 for an 8-bit image). We propose a histogram-based edge weight of nodes $t_{1},t_{2}$,
(8)$$\omega_{t_{1},t_{2}}=\max_{\begin{array}{c} k=1,\ldots ,L\\ m\in \{1,\ldots ,p_{r}\} \end{array}}L\left|\sum_{i=1}^{k}\left(f_{m,h_{i},t_{1}}-f_{m,h_{i},t_{2}}\right)\right|.$$The weight is in line with the histogram equalization technique that can dilute or remove the effects of disturbances, such as the effects of light intensity on images or videos. The method using the histogram-based edge weight is named HistEq-SHP.

## 3. Simulations

### 3.1. Comparative Study of the Performance of NonE-SHP and Maxima-SHP

We generated three-way tensor samples, $X_{t}\in R^{p_{1}\times p_{2}\times 3}$, using model (Equation 2). Under the null hypothesis, the model’s settings are $\mu_{m,1}=0,\;\sigma_{m,1}=1,\;\epsilon \overset{\mathrm{iid}}{∼}N\left(0,1\right)$ for m=1,2,3, and $d=p_{1}p_{2}=5$, 10, 100, 500, 1000, 5000, 10,000, 50,000. We carried out 10,000 simulations with α=0.05 and *T* = 20, 40, 60, 80, 100, 200. Table 1 displays the estimated type I errors via nonE-SHP and the proposed Maxima-SHP. All the estimated type I errors are around 0.05, consistent with the significance level α= 0.05.

Additionally, we examined the powers of Maxima-SHP and nonE-SHP under the alternative hypothesis. For simplicity, we also considered three-way tensor samples generated using model (Equation 2) with r=3. Suppose that $t^{*}$ is the preset change-point, and $K_{t}=\left\{1,3\right\}$ for $t>t^{*}$, indicating that the distribution of the first and third mode-3 slices $X_{1,t}$ change after $t^{*}$. When $t\leq t^{*}$, for the third mode-3 slice $\mu_{3,1}=0.2$, whereas for the first and second mode-3 slices $\mu_{1,1}=\mu_{2,1}=0$. When $t>t^{*}$, the distribution of the second mode-3 slice remains unchanged, whereas both distributions of the first and third mode-3 slices change, $\mu_{1,2}=0.2$ and $\mu_{3,2}=0$. Furthermore, let $\sigma_{m,1}=\sigma_{m,2}=1,\;\epsilon \overset{\mathrm{iid}}{∼}N\left(0,1\right)$ for m=1,2,3. Figure 4 visually shows the generated three-way tensors for $t\leq t^{*}$ and after $t^{*}$, i.e., $t>t^{*}$.

We performed the simulations via 10,000 repetitions at the significance level of 0.05. Figure 5 shows the estimated powers of Maxima-SHP and nonE-SHP by setting the change-point at T/2. Clearly, as the dimension *d* increases, the powers of nonE-SHP are all around 0.05, implying that it fails to reject the null hypothesis when the alternative hypothesis is true. However, the powers of Maxima-SHP tend to one for the whole *T*, empirically showing the strong power consistency. The power consistency of Maxima-SHP is mainly affected by the dimension *d*. Figure 6 shows similar results as Figure 5 when $\mu_{3,1}=\mu_{1,2}=0.3$ and keeping the other values unchanged. We conducted two more simulations by setting the location of change-point at T/4. The results are given in the Appendix B.

Additionally, we carried out other simulations under the scenario of unequal variances for different slices. The variance of the second mode-3 slice was equal to 1.5, whereas the other settings were the same as above. Table 2 shows that the estimated type I errors are still around 0.05 under the null hypothesis for both Maxima-SHP and nonE-SHP. Figure 7 gives the estimated powers under the alternative hypothesis. These results indicate that the proposed Maxima-SHP still outperforms the nonE-SHP, whether the variances of the slices are equal or not.

### 3.2. Comparative Study of the Performance of NonE-SHP and HistEq-SHP

In order to investigate the performance of HistEq-SHP, we first performed change-point detection by generating a sequence of synthetic gray images (*r* = 2) under different *T* and *d*. Figure 8 is an example of *T* = 40, *d* = 10,000, *r* = 2. In Figure 8, we depict images constructed using a normal distribution with a mean of 0.8 and a standard deviation of 0.05 for t=1,…,9 (see Figure 8a ), and using a normal distribution with a mean of 0.5 and a standard deviation of 0.05 for t=10,…,19 (see Figure 8b). Furthermore, we added an object, the shape of a butterfly, to the images constructed using the normal distribution with a mean of 0.5 and a standard deviation of 0.05 for t=20,…,29 (see Figure 8c). Finally, the illustration of the image in Figure 8d was generated on the basis of Figure 8a for t=30,…,40. The time point t=19 is the change-point. Under various *T* and *d*, we calculated the empirical powers based on 10,000 repetitions. We remark that we preprocessed the images using the histogram equalization technique before using the nonE-SHP method. Table 3 displays the powers of HistEq-SHP and nonE-SHP, which shows that the powers of both methods are ones when *d* = 5000 and 10,000 regardless of *T*. For *d* = 50,000, the powers of HistEq-SHP are ones, but those of the nonE-SHP method are close to one, implying the proposed method outperforms the nonE-SHP method for high dimension *d*.

We further examined the performance of these methods on three-way tensors by extending gray images to color images. As we discussed in the introduction, lighting can easily affect images, making change-point detection even more challenging. We performed another simulation to check the stability of the proposed methods by generating synthetic images under different light intensities. Figure 9 displays four types of images: Figure 9a is generated from a normal distribution with a mean of 0.8 and a standard deviation of 0.05 for three mode-3 slices, and a red object is added; Figure 9b is generated in the same way as in Figure 9a except that the mean of the normal distribution is 0.5; Figure 9c,d are generated in the same way as in Figure 9b and Figure 9a, respectively, except that blue targets replaced the red ones. Note that Figure 9a,d are in the same light intensity, and so are Figure 9b,c. It is obvious that the time change-point is at $t^{*}=19$. Table 4 gives the powers of nonE-SHP and HistEq-SHP. It shows that the powers of nonE-SHP are close to 0.05 in all cases, and, hence, the method failed to detect the real change-point, i.e., where the colors of the object are not the same, and the images receive different light intensities. On the contrary, the powers of HistEq-SHP are all ones. The results demonstrate that although the images are affected by different light intensities, the proposed HistEq-SHP method can successfully identify the true change-point.

## 4. Case Studies

### 4.1. Example 1: Color Change in a Traffic Light

Our proposed method has important implications in real life. An application that recognizes the color change of traffic lights may help daltonians reduce the limitations of their visual impairment and allow them to navigate traffic light intersections. We extracted 27 frames per second from a short video (27 s) taken at a traffic light intersection, recording the entire traffic light transition from green to red. We cropped each frame to size 73×40. Figure 1 displays three frames containing different colors of the traffic light, green (t=1,…,10), yellow (t=11,…,14), and red (t=15,…,27). Clearly, based on the hypothesis test in (Equation 3), two change-points exist: $t^{*}=10$ and $t^{*}=14$. In this case, we aim to detect the moments when the traffic light has changed its color, e.g., $t^{*}=10$ and 14.

We transformed the three-way tensor of the R,G,B slices into three vectors and normalized the pixel values. According to Equation (Equation 7), we extracted the maximum edge weights of the three color slices to construct SHP(ω). The test statistic $S_{T}^{\mathrm{SHP}(\omega )}$ in Equation (Equation 5) was 10.5787, larger than the corresponding critical value 2.1491 (T=27,d=2920), leading to the rejection of the null hypothesis at the significance level 0.05. The estimated location of the change-point was $t^{*}=14$. To further test if other change-points exist, we considered the two segments: $\left\{X_{t},t=1,\ldots ,13\right\}$ (Segment I) and $\left\{X_{t},t=15,\ldots ,27\right\}$ (Segment II). Following the same procedure as above, the second change-point was estimated at $t^{*}=10$ in Segment I, whereas there was no change-point in Segment II. The estimated results are given in Figure 10 (upper panel), which agrees with the real situation.

To compare Maxima-SHP with nonE-SHP, we also constructed the SHP(ω) via the non-Euclidean weight given by [2]. By following the same procedure, although the ratio cut was minimized at location 19, the statistic $S_{T}^{\mathrm{SHP}(\omega )}$ was 0.6278, less than the critical value, resulting in failing to reject the null hypothesis. The results are given in Figure 10 (lower panel). As expected, the nonE-SHP failed to detect the real change-points in such a scenario because it disregarded the main feature in three slices in constructing its non-Euclidean weights.

### 4.2. Example 2: Tomato Fruit Color Change

Nowadays, automatic harvesting systems have become a powerful technique for smart agriculture [7]. Monitoring the maturity stages of fruits or vegetables and estimating their harvest time are important steps in building an automatic harvesting system. Based on video recordings of fruit’s or vegetable’s maturation processes, the problem of monitoring the maturity stages and estimating the harvest times of fruit or vegetable can be transformed into a change-point detection problem.

Figure 2 shows three stages of the ripening process of the tomato fruit, MG (t=1,…,8), BR (t=9,…,19), and LR (t=20,…,26), containing 26 frames extracted from a video. Each of them was cropped to size 149 × 149. Obviously, $t^{*}=8$ and $t^{*}=19$ are considered as the change-points. Applying the change-point detection methods of HistEq-SHP and nonE-SHP, the results are displayed in Figure 11. As expected, both methods successfully detected the real change-points.

However, videos or images are easily affected by light intensity, which makes change-point detection more challenging. Here, we present another example to illustrate that our proposed method is more robust than the nonE-SHP when the recorded videos or frames are recorded in different light intensities. We darkened the frames of t=6,…,10 and lightened the frames of t=18,…,21, shown in Figure 3. Note that the change-points are still at $t^{*}=8$ and $t^{*}=19$, the detection of which is easily affected by the false change-points t=5, t=10, t=17, and t=21, where the light intensity changes. By applying nonE-SHP, three false change-points $t^{*}=5$, $t^{*}=10$, and $t^{*}=17$ were detected (see the lower panel of Figure 12). In contrast, the proposed HistEq-SHP detected two change-points at $t^{*}=8$ and $t^{*}=19$ (see the upper panel of Figure 12), consistent with the real ones. The nonE-SHP failed to detect the real change-points when the videos or frames were recorded in different light intensities. However, our proposed HistEq-SHP remained unaffected by disturbance(s) and successfully detected the real change-points.

## 5. Conclusions

Nowadays, videos or images are ubiquitously used in real life. This paper proposes a general class of graph-based change-point detection methods for checking whether objects with different characteristics and sequences of tensors are affected by natural disturbances. Simulation studies empirically showed the power consistency of the proposed methods. Furthermore, the proposed methods are shown to detect changes in the color of traffic lights at intersections and monitor changes in fruit ripening stages in smart farming. All numerical studies demonstrated the effectiveness of the proposed methods.

## Figures and Tables

**Figure 1 entropy-25-00552-f001:**
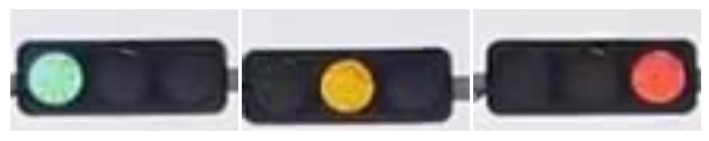
Three different colors of traffic lights: green (left panel, t=1,…,10), yellow (middle panel, t=11,…,14), and red (right panel, t=15,…,27).

**Figure 2 entropy-25-00552-f002:**
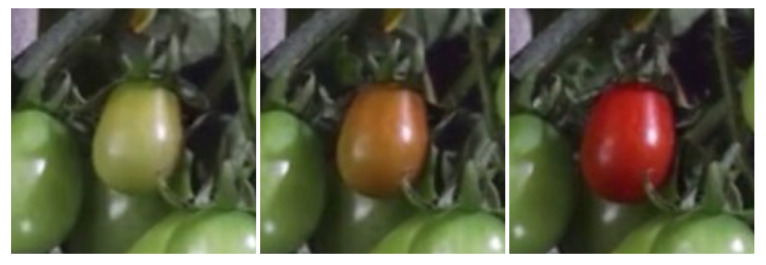
Three ripening stages: MG (left panel, t=1,…,8), BR (middle panel, t=9,…,19), and LR (right panel, t=20,…,26).

**Figure 3 entropy-25-00552-f003:**
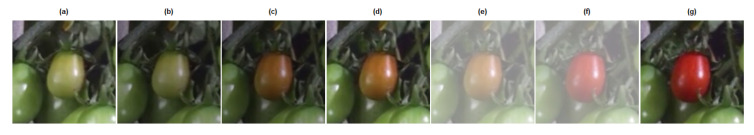
Three ripening stages where the light intensity changes at t=5,10,17,21: (**a**) t=1,…,5, (**b**) t=6,…,8, (**c**) t=9,10, (**d**) t=11,…,17, (**e**) t=18,19, (**f**) t=20,21, (**g**) t=22,…,26.

**Figure 4 entropy-25-00552-f004:**
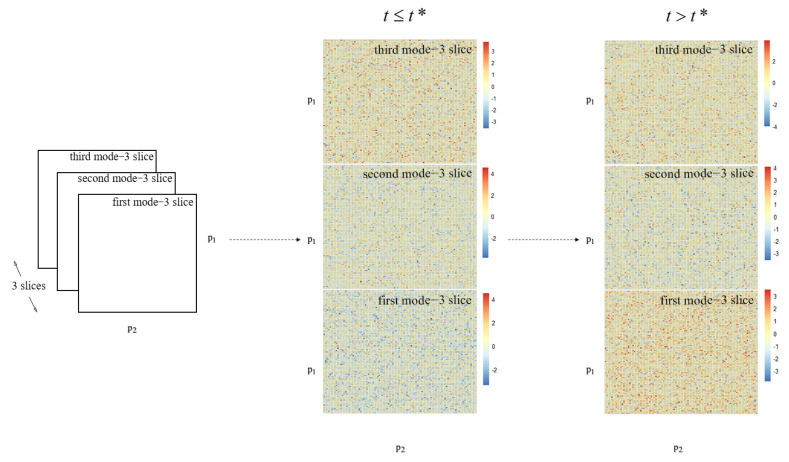
Illustration of the generated three-way tensors for $t\leq t^{*}$ and $t>t^{*}$ in the middle and on the right, respectively.

**Figure 5 entropy-25-00552-f005:**
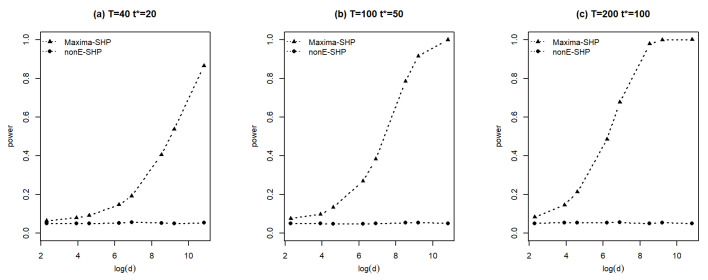
Estimated powers of Maxima-SHP and nonE-SHP over 10,000 simulations by setting equal slice variance; $t^{*}=T/2,\;T=40,100,200$, $\mu_{3,1}=0.2,\mu_{1,1}=\mu_{2,1}=0$ for $t\leq t^{*}$; and $K_{t}=\left\{1,3\right\}$, $\mu_{1,2}=0.2,\mu_{3,2}=0$ for $t>t^{*}$.

**Figure 6 entropy-25-00552-f006:**
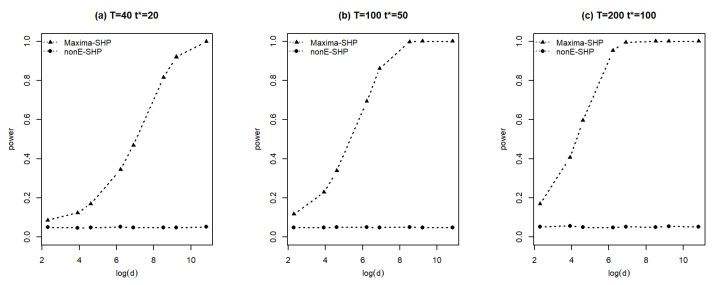
Estimated powers of Maxima-SHP and nonE-SHP over 10,000 simulations by setting $t^{*}=T/2,\;T=40,100,200$, $\mu_{3,1}=0.3,\mu_{1,1}=\mu_{2,1}=0$ for $t\leq t^{*}$; and $K_{t}=\left\{1,3\right\}$, $\mu_{1,2}=0.3,\mu_{3,2}=0$ for $t>t^{*}$.

**Figure 7 entropy-25-00552-f007:**
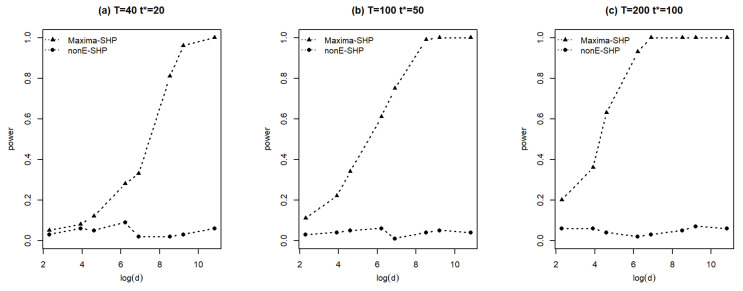
Estimated powers of Maxima-SHP and nonE-SHP over 10,000 simulations with unequal variances of slices by setting $t^{*}=T/2,\;T=40,100,200$, $\sigma_{2,1}=\sigma_{2,2}=1.5$, $\mu_{3,1}=0.3,\mu_{1,1}=\mu_{2,1}=0$ for $t\leq t^{*}$; and $K_{t}=\left\{1,3\right\}$, $\mu_{1,2}=0.3,\mu_{3,2}=0$ for $t>t^{*}$.

**Figure 8 entropy-25-00552-f008:**
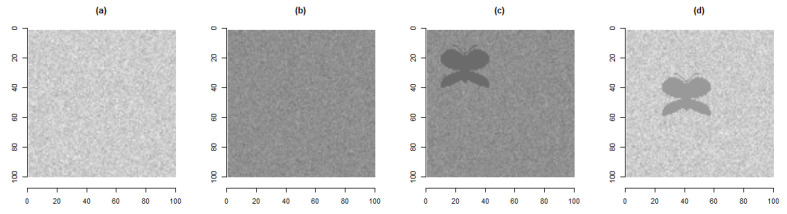
Illustrations of a sequence of synthetic gray images with *T* = 40, *d* = 10,000, r=2: (**a**) t=1,…,9; (**b**) t=10,…,19; (**c**) t=20,…,29; (**d**) t=30,…,40.

**Figure 9 entropy-25-00552-f009:**
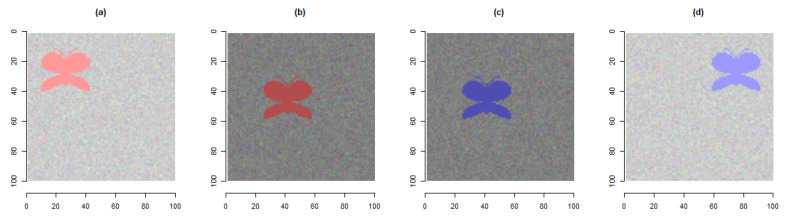
Illustrations of a sequence of synthetic color images with *T* = 40, *d* = 10,000, *r* = 3: (**a**) t=1,…,9; (**b**) t=10,…,19; (**c**) t=20,…,29; (**d**) t=30,…,40. (**a**,**d**) are in the same light intensity, and so are (**b**,**c**).

**Figure 10 entropy-25-00552-f010:**
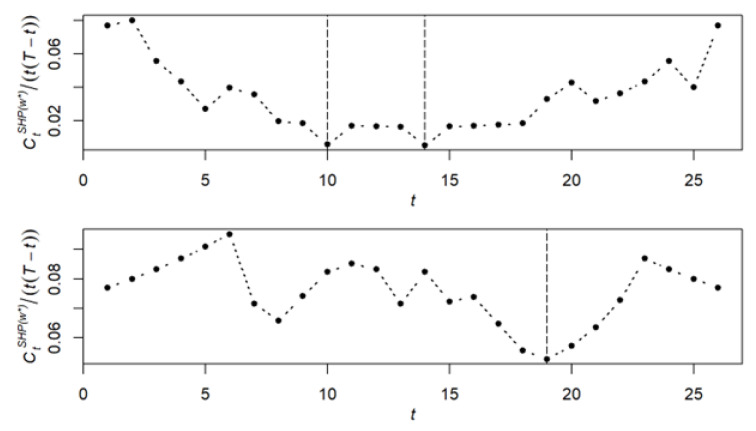
The change-point estimation using Maxima-SHP (**upper panel**) and nonE-SHP (**lower panel**).

**Figure 11 entropy-25-00552-f011:**
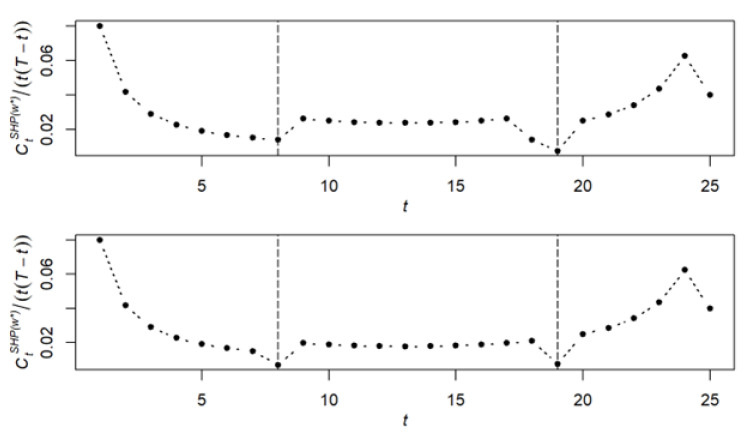
Change-point detection via HistEq-SHP (**upper panel**) and nonE-SHP (**lower panel**).

**Figure 12 entropy-25-00552-f012:**
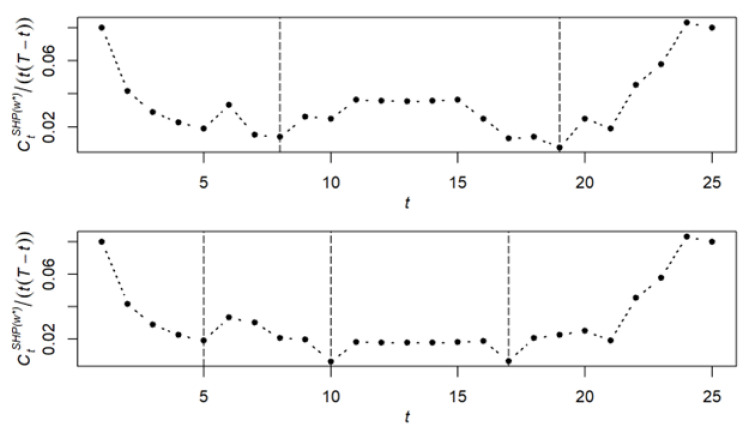
Change-points detected by HistEq-SHP (**upper panel**) and nonE-SHP (**lower panel**), light intensity changes at t=5,10,17,21.

**Table 1 entropy-25-00552-t001:** Comparison of type I error rates using nonE-SHP and Maxima-SHP based on various *T* and *d*.

*T*	Method	*d*							
		10	50	100	500	1000	5000	10,000	50,000
20	Maxima-SHP	0.0519	0.0505	0.0493	0.0495	0.0524	0.0495	0.0507	0.0513
	nonE-SHP	0.0446	0.0506	0.0502	0.0510	0.0535	0.0494	0.0496	0.0463
40	Maxima-SHP	0.0512	0.0495	0.0519	0.0500	0.0561	0.0503	0.0460	0.0531
	nonE-SHP	0.0529	0.0498	0.0484	0.0496	0.0478	0.0530	0.0502	0.0530
60	Maxima-SHP	0.0507	0.0491	0.0479	0.0508	0.0483	0.0477	0.0497	0.0497
	nonE-SHP	0.0487	0.0491	0.0489	0.0518	0.0460	0.0504	0.0507	0.0469
80	Maxima-SHP	0.0487	0.0495	0.0508	0.0479	0.0565	0.0468	0.0505	0.0514
	nonE-SHP	0.0522	0.0513	0.0493	0.0538	0.0522	0.0480	0.0519	0.0500
100	Maxima-SHP	0.0459	0.0498	0.0465	0.0476	0.0517	0.0481	0.0484	0.0484
	nonE-SHP	0.0522	0.0506	0.0521	0.0458	0.0497	0.0456	0.0496	0.0483
200	Maxima-SHP	0.0456	0.0507	0.0480	0.0514	0.0510	0.0513	0.0500	0.0540
	nonE-SHP	0.0476	0.0514	0.0473	0.0489	0.0466	0.0519	0.0434	0.0503

**Table 2 entropy-25-00552-t002:** Comparison of type I error rates of nonE-SHP and Maxima-SHP with unequal variances of slices, based on various *T* and *d*.

*T*	Method	*d*							
		10	50	100	500	1000	5000	10,000	50,000
20	Maxima-SHP	0.0532	0.0479	0.0537	0.0484	0.0464	0.0520	0.0516	0.0497
	nonE-SHP	0.0489	0.0468	0.0533	0.0471	0.0511	0.0525	0.0494	0.0468
40	Maxima-SHP	0.0475	0.0519	0.0501	0.0473	0.0516	0.0464	0.0533	0.0525
	nonE-SHP	0.0490	0.0495	0.0542	0.0478	0.0529	0.0500	0.0498	0.0527
60	Maxima-SHP	0.0518	0.0531	0.0496	0.0492	0.0476	0.0508	0.0532	0.0521
	nonE-SHP	0.0469	0.0471	0.0456	0.0514	0.0538	0.0464	0.0565	0.0487
80	Maxima-SHP	0.0485	0.0493	0.0462	0.0455	0.0530	0.0472	0.0504	0.0521
	nonE-SHP	0.0498	0.0476	0.0446	0.0487	0.0501	0.0481	0.0466	0.0504
100	Maxima-SHP	0.0495	0.0472	0.0514	0.0508	0.0534	0.0494	0.0530	0.0503
	nonE-SHP	0.0505	0.0496	0.0483	0.0441	0.0513	0.0504	0.0477	0.0534
200	Maxima-SHP	0.0489	0.0527	0.0537	0.0502	0.0521	0.0479	0.0543	0.0487
	nonE-SHP	0.0461	0.0528	0.0516	0.0523	0.0515	0.0510	0.0501	0.0462

**Table 3 entropy-25-00552-t003:** Comparison of the powers of nonE-SHP and HistEq-SHP based on gray images with various *T* and *d*.

*T*	Method	*d*		
		5000	10,000	50,000
40	nonE-SHP	1.0000	1.0000	0.9110
	HistEq-SHP	1.0000	1.0000	1.0000
100	nonE-SHP	1.0000	1.0000	0.9991
	HistEq-SHP	1.0000	1.0000	1.0000

**Table 4 entropy-25-00552-t004:** Comparison of the powers of nonE-SHP and HistEq-SHP based on color images with various *T* and *d*.

*T*	Method	*d*		
		5000	10,000	50,000
40	nonE-SHP	0.0503	0.0485	0.0499
	HistEq-SHP	1.0000	1.0000	1.0000
100	nonE-SHP	0.0527	0.0507	0.0494
	HistEq-SHP	1.0000	1.0000	1.0000

## Data Availability

Not applicable.

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
