# Peer review of "Change-Point Detection for Multi-Way Tensor-Based Frameworks"

_entropy, 2023, doi:10.3390/e25040552_

Round 1
Reviewer 2 Report
The paper builds heavily on the the works by Shi et al. (2017, 2018). There are two extensions that the authors pursue. First, rather than try to detect a change using the average all modes, the paper treats each mode separately so that a change in an individual mode can be better detected (e.g., finding that the red slice in an RGB image changed). Second, the authors consider the case when mean intensity levels of tensor slices change as a result of differences in, e.g., lighting conditions, which may degrade the ability to detect object changes.
I have two concerns:
(i) The proposed methods seem like straightforward extensions of Shi et al. For example, taking the max in equation 7 over R,G,B slices instead first averaging the slices is a minor change. Perhaps the authors can better clarify in the text what is novel and interesting from a modeling/math point of view.
(ii) Appropriate benchmarks are not included. For example, Shi et al. methods were not designed to account for when the lighting intensity changes. Therefore, it is completely expected that the proposed method would do better than the original Shi et al. methods for this case. But there are many alternative approaches that do account for lighting intensity changes, which would serve as more appropriate benchmarks. The authors should show how their method performs against other methods to better establish their contribution against the state-of-the-art.
Round 2
Reviewer 1 Report
The authors have addressed my comments well.
Author Response
Thank you very much for your helpful suggestions.
Reviewer 2 Report
Thanks to the authors for their responses and updates to the manuscript.
I remain of the opinion that the novelty of the paper is low. This is fine as long as there is an improvement in performance over benchmark methods. However, the paper still does not compare against an appropriate benchmark. The paper does make comparisons with Shi et al, but Shi et al is not actually not designed for color images. As such, any reader would not know when to use the proposed method or an alternative technique from other scholars.
Round 3
Reviewer 2 Report
no further comments.
Author Response
We have made a great effort in correcting spelling errors and grammar using Grammarly premium.